**Data Availability Statement:** As the participants were relatively small number of health professionals, in a specific location, data contains

# Mental health service utilization in a low resource setting: A qualitative study on perspectives of health professionals in Northwest Ethiopia

**Tesfa Mekonen**[1,2,3]*, **Gary C. K. Chan**[2], **Tilahun Belete**[3], **Melak Menberu**[3], **Lily Davidson**[1], **Leanne Hides**[1,2], **Janni Leung**[1,2]

1 School of Psychology, The University of Queensland, Brisbane, Australia, 2 National Centre for Youth Substance Use Research, The University of Queensland, Brisbane, Australia, 3 Psychiatry Department, Bahir Dar University, Bahir Dar, Ethiopia

* t.yimer@uq.net.au

## Abstract

### Background

Mental health services are advocated to move from specialized care to more integrated and accessible primary care settings. The integration of mental health into primary health services is expanding in Ethiopia. However, there is a lack of research on the perspectives of health professionals on mental health services in Ethiopia. This study aimed to explore the perspectives, views, and experiences of health professionals regarding mental health services and the help-seeking behavior of people with mental illness in Northwest Ethiopia.

### Methods

This qualitative study included sixteen health professionals. The participants were interviewed using a semi-structured interview guide. Interviews were conducted in Amharic and translated into English. We analyzed the data using reflexive thematic analysis.

### Results

Five key themes were generated from the interview data: (1) Avoidance of mental healthcare; (2) Low literacy about mental illness; (3) Stigma as a hindrance to accessing mental health services; (4) The role of culture and religion in mental health services; and (5) Lack of government concern about mental health services.

### Conclusion

This study identified important factors related to mental health services in Northwest Ethiopia, from the perspectives of health professionals. Our findings highlighted that the government health agenda needs to prioritize mental health services. The role of religious and

potentially sensitive information to share it publicly. Data can be requested from Human research ethics committee, The University of Queensland (humanethics@research.uq.edu.au) or Institutional review board, College of Medicine and Health Sciences, Bahir Dar University (infocmhs@bdu.edu.et).

**Funding:** This research did not receive any specific grant from funding agencies. The data collection cost was covered by College of Medicine and Health Sciences, Bahir Dar University, Ethiopia. TM is supported by The University of Queensland Research and Training Program (RTP) scholarship. The funder had no role in the study design, data collection and analysis, decision to publish, or preparation of the manuscript.

**Competing interests:** The authors have declared that no competing interests exist.

cultural healing places in the help-seeking pathway should be given due consideration to ensure success in mental health services provision and utilization.

## Background

According to the 2019 global burden of disease estimate, more than 970 million people live with mental illness which indicates a 48.1% increase since 1990. Of all people with mental illness, 66.7% reside in low and middle SDI (Sustainable Development Index) countries [1]. Due to the COVID-19 pandemic, mental illness has posed a great concern with a 25% worldwide increase in the prevalence of depression and anxiety [2]. In Ethiopia, a 3-fold increase in the prevalence of depression was reported compared to the pre-pandemic estimate [3]. Despite the disproportionate burden of mental illness, mental health resources are scarce in low- and middle-income countries (LMIC) [4], which creates a massive treatment gap [5, 6].

People with mental illness often experience greater difficulty accessing mental health services due to different attitudinal and structural barriers [7]. To narrow the treatment gap, many countries adopted the initiative of integrating mental health services into primary healthcare [8, 9]. Despite the efforts to increase the accessibility of mental health services, treatment rates for mental illness remain low in LMIC. Systematic reviews and meta-analyses have found that only 9.3% of people with alcohol use disorder [5] and 16.8% of people with depression [6] access treatment in their lifetime in low-income countries. The low treatment rate maybe partly due to the integration being hindered by numerous barriers including limited resources and low mental healthcare needs from the public [10, 11].

As mental health services move from specialized to integrated primary healthcare services, it is important to explore the perspectives of health professionals on mental health services. The integration of mental health and primary health services is also expanding in Ethiopia [12] to help achieve equitable access for different segments of the population [13]. However, there is a lack of research on the perspectives of health professionals on mental health services in Ethiopia and potential barriers to accessing them to help explain the large treatment gap.

Critical to improving mental health services is informing policymakers the potential challenges in mental healthcare provision. Understanding the perspectives of health professionals can help plan strategies to improve mental health service provision and utilization. This qualitative study aimed to explore the perspectives, views, and experiences of health professionals regarding mental health services and the help-seeking behavior of people with mental illness in Northwest Ethiopia.

## Methods

### Setting

The Ethiopian healthcare system is organized in three-tier systems including primary, secondary, and tertiary levels (Fig 1) [14]. The primary healthcare comprises a primary hospital, a health center, and five satellite health posts (the lowest level health system facility at the village level). Health posts are expected to provide basic infectious disease and maternal and child health care. A health center serves as a referral center for health posts and is expected to provide both preventive and curative services for people with any illness. A primary hospital serves as a referral center for health centers and provides inpatient and ambulatory services, including emergency surgical services. A general hospital is a referral center for primary hospitals. Similarly, a specialized hospital is a referral center for general hospitals. This ladder of referral

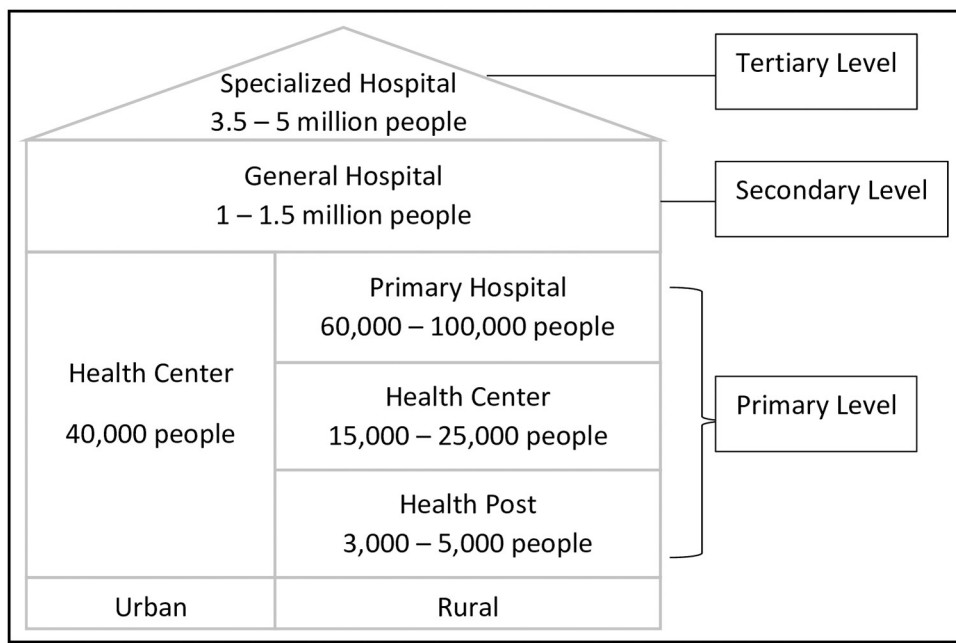

**Fig 1. Ethiopian healthcare tier system (source: Health sector development program IV [14])–this diagram is for any illness, not specific for mental illness.**

system is for all patients with any illness. There is only one dedicated psychiatric hospital (268 beds) and one general hospital with a large share of psychiatric beds (150 psychiatric beds) in Ethiopia, both located in Addis Ababa [12]. This qualitative study was conducted between March and July 2021 in two specialized hospitals, two primary hospitals, and four health centers, located in Northwest Ethiopia.

## Sampling strategy and participants

Two authors (TB and MM) recruited the participants by providing information about the study in the health facilities and searching for potential participants. Participants were key informants (KI) selected purposively to include health professionals working in different tiers of the Ethiopian healthcare system. Participants were approached in person and asked if they were willing to participate in the study. Those participants who expressed interest were informed about the study aims and interview procedure.

A total of 17 participants were approached and 16 (5 female,11 male) participants have participated in this study. One participant (male) agreed to be involved in the study, but later declined the interview due to time constraints. Participants' ages ranged from 25–47 years. Six participants were from health centers, four were from primary hospitals, and six were from specialized hospitals. In recruiting participants, we also considered professional diversity (e.g., physicians, nurses, and other healthcare providers) and roles in the healthcare system (e.g., clinical or leadership roles). Regarding professional diversity, participants were four nurses (two B.Sc. in nursing and two B.Sc. In psychiatric nursing), three general practitioners (Medical Doctors), three public health officers (B.Sc. in public health), one psychiatrist, one mental health specialist (M.Sc. in mental health), two specialist physicians (surgeon and neurologist), and two other health professionals (M.Sc. [specialization not specified]). Of these participants, three were in leadership positions in the health facility, and all others occupied clinician roles. Other than mental health professionals, most of the participants had a brief experience of

providing mental healthcare and three participants never had experience providing mental healthcare.

## Ethical considerations

Written informed consent was obtained from each participant. Ethical clearance was obtained from The University of Queensland (approval number: 2020/HE000945) and College of Medicine and Health Sciences, Bahir Dar University (approval number: 009/2021).

## Data collection/interview technique

Two authors (TB and MM) collected the data using a semi-structured interview guide and a digital audio recorder. The interview guide contained questions about participants' characteristics and their views on mental health services in Ethiopia, people's help-seeking behavior for mental health issues, and potential barriers to psychiatric patients seeking professional help (see S1 File). The interview took place in the participant's office or another convenient location within the health facility. The length of the interviews ranged from 20 to 55 minutes.

## Data processing and analysis

Data analysis (including transcription) was conducted by the first author (TM) concurrently with the data collection from May 2021 to February 2022. The data were analyzed using a reflexive thematic analysis approach [15], which involves an iterative process of reading, coding, and theme development in each transcript to ensure a rich and detailed analysis of patterns across the dataset [16]. The analysis was theoretically informed by critical realism, an approach that recognizes reality as being constructed from a person's perspectives and experiences of social constructs including culture, language, and political interest [17]. As such, we treated the participants' responses as 'real' experiences and interpreted them based on socially available implications. As outlined by Braun and Clarke [15, 18], the analysis process involved six steps:

1. Familiarization with the data: The first author (TM) reviewed each interview by listening to the audio recording and transcribing it manually. After completing the transcription, each script was read at least twice.

2. Coding: The transcribed data were organized and coded using NVivo 12 software for qualitative data analysis [19]. All interview transcripts were coded using an inductive approach, which allows the data to guide the generation of themes [20]. During the coding stage, each section of text was analyzed, and relevant codes were generated to capture the meaning of each text section. Codes were generated and applied throughout the full transcript. After coding each transcript, the first author re-read each transcript to become familiar with the initial codes, to inform the next step.

3. Generating initial themes: To generate initial themes, patterns of codes were identified by reading and cross-checking the codes against the data.

4. Reviewing themes: Initial themes generated in Step 3 were reviewed by cross-checking them against the data and revising themes as necessary. Initial themes with similar meanings were merged to generate final themes (see Table 1).

5. Defining and naming themes: The 'final themes' generated in step 4 were named and given definitions (see Table 1).

**Table 1. Themes with description.**

| | Themes | Theme description |
|---|---|---|
| i | Avoidance of mental healthcare | In the view of health professionals, mental health service is time-consuming and a secondary consideration to physical illness. Psychiatric patients are not given equal priority as patients with physical illness. Psychiatric patients are being abandoned by their families, the community, and health professionals. |
| ii | Low literacy about mental illness | Mental illness is not taken seriously in the community and the health facilities. Having a mental illness is perceived as a sign of failure or laziness. The lack of awareness is common among the patients, communities, and health professionals. |
| iii | Stigma as a hindrance to accessing mental health services | People avoid seeking treatment due to the stigma/discrimination associated with mental illness diagnoses. |
| iv | The role of culture and religion in mental health services | Religion and culture play a significant role in mental health services. People usually prefer to seek help from religious/cultural places. Health care settings are the last resort in the help-seeking pathway, creating an extended period of untreated mental illness. |
| v | Lack of government concern about mental health services | Mental health is not a priority in Ethiopia's current public health policy. Government policies are more inclined to physical health problems. |

6. Writing-up: The reporting of this research follows the guideline provided by the Standards for Reporting Qualitative Research (SRQR) checklist (see S2 File) [21].

## Results

Five themes were generated from the interview data provided by the 16 participants/key informants (KI) (see Table 1).

### 1. Avoidance of mental healthcare

A prominent theme generated from the data was an intention to avoid providing mental healthcare. In the Ethiopian healthcare system, health professionals in primary care facilities are the frontline workers in the healthcare-seeking pathway for people with any illness, including mental illness. However, most participants did not consider mental health services to be a primary responsibility. In these busy primary care facilities (health centers), mental health service is perceived as both time-consuming and a secondary consideration to physical illness. Some participants explicitly expressed that "they have no time" to provide care for mentally ill patients as health professionals are busy providing care to patients with physical illnesses.

*Because every professional is serving 50 or more [non-psychiatric patients] per day, it is hard in this crowded time to talk [to psychiatric patients] and to give appointments for follow-up.*

*(KI 5)*

*Considering the high patient load [of non-psychiatric patients], it is difficult to provide care for psychiatric patients. There are time constraints.*

*(KI 6)*

Health centers are perceived not to be the appropriate place to treat mental illness. This idea was exemplified by some of the participants working in the health centers (mostly nurses and public health officers), such as: "*we just refer them [to hospitals], we don't initiate treatment*" [KI 8], and "*care providers tend to push their responsibility away*" [KI 10]. Contrary to

participants working in the health centers, participants working in hospitals (mostly mental health professionals and specialist doctors) implied that people with mental illness should be treated at health centers (e.g., "*patients that supposed to be treated at health centers are referred to specialized hospitals which creates undue burden*" [KI 3]). Participants' references to care providers 'referring on' or 'pushing away' patients with mental illness exemplify the common theme of avoidance of mental healthcare.

> *It is difficult to treat someone with mental illness here in the health center. we just provide follow-up treatment after they've become stable. Usually, we directly refer them to hospitals, we do not give treatment here in the health center.*
>
> *(KI 9)*

Most health professionals held the perception that mental illness is not treatable in health centers or not treatable at all. One participant described how this sentiment is also held by people in the community, even those who are well educated.

> *I guess many of them [health professionals] don't think mental illness is treatable. I don't feel like they think that patients with mental illness will improve with treatment.*
>
> *(KI 4,)*

> *Forget the rural community, even the educated people, those who work in the medical field, have no good attitude about the treatment and if it [mental illness] is treatable at all.*
>
> *(KI 3)*

Despite the general tendency of health professionals to avoid providing mental healthcare, some participants who had on-the-job training on mental health reported more engagement in providing mental health services (e.g., "*I'm providing mental health treatment with full confidence after receiving the on-the-job training*" [KI 2]). Some participants also reported their motivation to engage in mental health services if they can be trained in how to manage people with mental illness. Lack of knowledge and skill in diagnosing and treating mental illness was one of the reasons mentioned by participants as to why health professionals tend to avoid providing mental health services (e.g., "*I don't have the training to give mental healthcare*" [KI 6]; "*Mental health training is necessary, especially for primary care professionals as they are in the front line*" [KI 5]). Some of the participants also reported that mental health training/course given to graduates in the University is not enough to provide care for patients (e.g., "*I cannot have the courage to manage the case [mental illness] just by the knowledge I get from the university*" [KI 2]; "*I don't think health professionals can handle a patient just by the course they've taken at the university*" [KI 4]).

## 2. Low literacy about mental illness

Another prominent theme generated across most interviews was low literacy about mental illness. Most participants reported a low level of mental health literacy in the community, including among health professionals (e.g., "*Let alone other people, even health professionals don't have much knowledge about mental illness*" [KI 1]; "*Whether it is in the community or health professional, the awareness about mental health is very low*" [KI 4]).

Low mental health literacy was apparent in participants' report of misconceptions about the cause, severity, and treatment of mental illness. According to some participants, the general community perceived the cause of mental illness to be "*an evil spirit* [KI 5]", "Satan or

*Supernatural forces* [KI 14]*"* or that mental illness is a *"spiritual illness* [KI 4]*"*. The severity of mental illness was downplayed by health professionals and members of the broader community (e.g., *"those higher professional officials, they do not give attention to mental illness because it is not fatal"* [KI 4]). Some participants reported that mental illness may not be considered to be an illness at all (e.g., *"In the community, a person is considered ill when he/she has some sort of physical illness. Mental illness has never been seen as illness"* [KI 15]), depression is particularly downplayed as a *"sign of laziness* [KI 11]*"* or *"part of day-to-day life* [KI 5]*"*.

These common misconceptions about the cause and severity of mental illness lead people to think that the problem is not important enough to seek professional help (e.g., *"patients do not come [to health facilities] by themselves. They usually come when they think the problem is big enough, like epilepsy"* [KI 1]), or they do not come at all, thinking mental illness is not treatable (e.g., *"many community members are not sure if it [mental illness] is treatable"* [KI 3]). Some participants also reported that many health professionals held the misconception that mental illness is 'not treatable' (e.g., *"many health professionals do not think mental illness is treatable"* [KI 4]). It is apparent that mental health literacy is low among health professionals as well as the border community.

### 3. Stigma as a hindrance to accessing mental health services

People avoid seeking professional help for mental illness, for fear of experiencing stigma and discrimination (e.g., *"They [patients] faced so much pressure from the community, . . . they fear being discriminated, so they prefer to solve their problem by themselves"* [KI 5]). People with mental illness tend to avoid being seen in mental health units or by mental health professionals even if they want treatment (e.g., *"we have epilepsy patients who also have some psychiatric illness. They do not want to go to the psychiatric OPD [outpatient department], they want us to do the medication refill here. They want the medication, but they do not want to be seen in psychiatric OPD* [KI 12]*"*).

The societal implications of receiving a mental illness diagnosis have a large impact on mental health service utilization. Because of the stigma and discrimination from the community, families put the patients behind a locked door, isolating them from the outside world.

> *The families put them [patients] in torture, as a prisoner, locked behind so that no one can see them.*
>
> *(KI 10)*

> *You know, if someone is known as mentally ill or as crazy, no one will give him a job or will never get married.*
>
> *(KI 15)*

In addition, clients *"refuse to accept mental illness diagnoses* [KI 12]*"* and health professionals did not feel comfortable telling the diagnosis directly (e.g., *"Actually, I also didn't tell [the diagnosis] directly to them [patients]"* [KI 12]), which creates a missed opportunity to treatment and limits the progress to mental health promotion.

### 4. The role of culture and religion in mental health services

According to the participants, people view seeking help from health facilities for mental health problems as a last resort. If people do seek help for mental health problems, their first choice of help is usually a religious place and traditional healer (e.g., *"If a person showed such a strange*

*behavior, they would take him to the holy water, not to the health facility"* [KI 13]). Similarly, participants reported on the role of religious practices as a preference in treating mental health problems, compared to health facilities (e.g., *"almost all psychiatric patients try holy water before coming to a health facility; they come after they give up"* [KI 15]). According to one participant, symptoms of mental illness are usually attributed to other 'socially acceptable' norms, such as supernatural or religious phenomena. For instance, it is more socially acceptable to refer to a person's mental illness as a *"possession* [KI 15,]*"* because of the stigma and discrimination attached to being diagnosed with a mental illness.

> *It is more acceptable in the community to call it [mental illness] 'possession' or something socially acceptable.*
>
> *(KI 15)*

The tendency to seek help from places of worship relates to beliefs about the cause of mental illness. People prefer to seek help from religious and traditional places over health facilities because they believe their illness is caused by Satan or some Supernatural force (e.g., *"in our community, it is believed that mental illness is a spiritual illness. So, where would they go first before they come to us? It is religious treatment"* [KI 4]). In addition, going to religious places for mental health reasons is perceived as normal, contrary to seeking treatment in health facilities (e.g., *"forget hospitals, they[patients] even do not want to go to holy water (KI 13)"*. In this quote, the participant implied that going to holy water for mental illness treatment is more normal than going to hospitals, but patients sometimes refuse even this option.

Participants commented on the challenging nature of engaging people in mental health service utilization when the community prefers to seek help from traditional and religious places over the health facilities (e.g., *"I don't think our culture is encouraging for medical treatment. You can get more patients in holy water and traditional places than hospitals"* [KI 12]). However, the participants also recognized the opportunity to access high numbers of patients by integrating mental health services into traditional and religious places (e.g., *". . .if the religious leaders have the awareness, it is possible to refer the patient to us while they are also taking the religious treatment. I think the religious leaders could help a lot if they get the knowledge"* [KI 4]).

## 5. Lack of government concern about mental health services

Another prominent theme was the low priority given to mental health in the Ethiopian health policy. All of the participants who had experience of providing mental healthcare reported the lack of government concern in the Ethiopian mental health services. According to some participants, it is difficult to address the mental health problem with the *"health professionals' capacity* [KI 1]*"* as *"the government never considered mental health as a health agenda* [KI 1]*"*. As there are no public or private funds focusing on mental health, *"the policies are more inclined towards the diseases that have more external funds* [KI 12]*"*.

The loose government involvement in the mental health system is reflected by the lack of infrastructure and systematic sidelining of mental health in public health policies (e.g., *"I would say, the health policy is more focused on communicable diseases . . . mental health is especially ignored"* [KI 13]). This lack of government concern in the mental health system creates a scarcity of mental health services and uncertainty in the health professionals' perceived preparedness to provide service for those who are in need.

> *Sometimes we say that "it will be great if we go out and mobilize the community to come to a health center for mental health issues", but what if they all come? How can we manage them?*

*We can't! In this room, we do cervical cancer screening, diabetes mellites, hypertension, and mental health services. And the trained health professionals are just two.*

*(KI 4)*

The lack of government concern for mental health leads to significant structural barriers to accessing mental health services. The long-distance that should be traveled to get treatment is *"preventing the patient from coming to health facilities [KI 4]"*. Lack of trained manpower and lack of mental health infrastructures in the general health facilities creates a missed opportunity (e.g., *"we don't have a psychiatric ward to admit the patient, we don't even have a doctor who has the proper training [psychiatrist] and who could understand the patient"* [KI 5]). Participants suggested that the common perception that mental illness is 'not fatal' contributes to the low priority given to mental health in the government's health agenda.

*The government is focusing on infectious diseases only. People with mental illness are ignored and forgotten. They have no one to remember them and they are despised. There is no concern, it is visible that they are neglected.*

*(KI 4)*

## Discussion

This study explored health professionals' views, perspectives, and experiences on mental health services in Northwest Ethiopia and the help-seeking behavior of people with mental illness. From the perspective of health professionals, lack of government concern, low mental health literacy, stigma about mental illness, and cultural and religious practices are important factors to the accessibility and utilization of mental health services. Compounded with these factors, health professionals appear to avoid providing mental healthcare, and people with mental illness tend to avoid seeking mental healthcare.

Several misconceptions about the cause, severity, and treatment of mental illness affect the accessibility and utilization of mental health services. Perceptions that mental illness is caused by supernatural forces prevent many people from seeking help. The misconception that mental illness is non-fatal appears to diminish the perceived urgency to prioritize mental health in the government health agenda. However, with approximately 8 million deaths each year, mental illness is among the most substantial causes of death globally [22]. Moreover, the comorbidity of mental illness with physical illness is also common, which leads to worsening of the prognosis and increased treatment costs [23]. Mental illness, such as depression is also the leading cause of years lived with disability in Ethiopia [24].

Recent developments in the Ethiopian mental health system are promising as evidenced by the launch of the national mental health strategy [12, 25] and the efforts of the World Health Organization mental health gap action program to increase access to mental health services in the general healthcare [9]. However, compared to the health and economic burden caused by mental illness in Ethiopia [26–28], the availability and accessibility of services remain very limited. Our results suggest that the government health agenda is not giving adequate attention to increase access to mental health services in Ethiopia. As one of the prominent themes was low mental health literacy among healthcare professionals and the general public, government commitment should prioritize to increase mental health literacy. Our result highlighted that mental health courses taught at universities do not adequately prepare graduates to provide mental health services. Medical students in Ethiopia spend approximately eight weeks on theoretical training and two to four weeks on practical training in mental health. Other health

sciences students (e.g., Midwives and Nurses) spend approximately four weeks on theoretical and two weeks on practical training in mental health. The literacy among health professionals can be improved through better incorporation of mental health education in their university training and by providing on-the-job training [29, 30]. Consistent and large-scale campaigns [31] are needed to increase mental health literacy among the general community.

There is a missed opportunity that mental illnesses such as depression is left undetected for those who did access primary care services as reported by a recent study where only 7% of depression was detected by primary care clinicians in LMIC [32]. Previous study in Ethiopia also reported that 98% of depression cases were undetected by primary care clinicians [33]. To improve mental health services and utilization, the attitude [11] of health professionals towards providing mental healthcare should be addressed. Consistent with the results of the current study, previous studies reported the common perceptions among primary care clinicians that primary care facilities are not suited to treat mental illness and that specialists should treat mental illness [34, 35]. Our study also found that health professionals referred clients seeking treatment for common mental illnesses to specialized hospitals based on the perception that mental healthcare is demanding specialist skills. Referring clients to specialized hospitals for a condition that could be handled at the primary care level creates an undue burden in specialized hospitals. To improve health professionals' motivation [36] to provide mental health services, education and on-the-job training should be provided.

Traditional and religious places play significant roles in the help-seeking pathway of people with mental illness. Our study indicated that most people with mental illness seek help from religious and traditional places before visiting healthcare facilities. A previous Ethiopian study also reported consistent finding that 88% of patients with mental illness visited religious and traditional places before seeking professional help [37]. A promising avenue for increasing mental health service utilization may be to utilize the existing help-seeking pathway of religious and traditional healing places, either by providing basic mental healthcare training to key members of these places (e.g., religious leaders and traditional healers), or to encourage and educate these key members to increase referrals to health facilities. There are also reports from previous studies in Ethiopia that patients and family members expressed comfort in taking both pharmacological and religious/traditional treatments at the same time [37, 38]. As such, promoting mental health services without due consideration of the importance of religious and traditional places in the help-seeking pathway may not be an effective approach.

Interpretation of this finding should consider the following two issues: First, as we used Braun and Clarke's reflexive thematic analysis approach [15, 18], the analysis was conducted by one author. We are aware of other thematic analysis approaches such as the coding reliability approach (the analysis requires multiple coders to check for reliability) [39]. However, as the preferred analysis for this study was reflexive thematic analysis, a one researcher analysis is the proper approach [20]. Second, this study was limited by a small sample (n = 16) and lack of variability among participants. We focused only on health professionals' views and perspectives about mental healthcare provision and help-seeking in Northwest Ethiopia. To have a comprehensive picture of mental health service provision and utilization, future studies need to interview policymakers and clients with mental illness. For instance, interviewing policymakers would provide useful insights into the reasons for the lack of public and private funding for mental health. Attitudinal barriers to seeking mental health services could be better explored from the patient's perspective.

## Conclusion

Despite the recent efforts to decentralize mental health services in Ethiopia, the perception that mental illness should be treated in specialized hospitals is still held by many health

professionals. Moreover, the low literacy on mental health, stigma towards mental illness, and lack of government concern are major barriers to mental health service provision and utilization. Mental health education is needed in healthcare settings, schools, and the wider community. Mental health training and capacity building for health professionals can help to improve mental health services. The government health agenda needs to prioritize mental health services by providing an adequate resource for mental healthcare. The role of religious and cultural healing places in the help-seeking pathway should be given due consideration to ensure success in mental health services provision and utilization.

## Supporting information

**S1 File. Interview guide.**
(DOCX)

**S2 File. Standards for Reporting Qualitative Research (SRQR) checklist.**
(DOCX)

## Acknowledgments

We would like to thank the study participants for their involvement in this research.

## Author Contributions

**Conceptualization:** Tesfa Mekonen, Gary C. K. Chan, Leanne Hides, Janni Leung.

**Data curation:** Tesfa Mekonen, Tilahun Belete, Melak Menberu.

**Formal analysis:** Tesfa Mekonen.

**Funding acquisition:** Tesfa Mekonen.

**Investigation:** Tesfa Mekonen, Tilahun Belete, Melak Menberu, Leanne Hides, Janni Leung.

**Methodology:** Tesfa Mekonen, Janni Leung.

**Project administration:** Tesfa Mekonen, Tilahun Belete, Melak Menberu, Janni Leung.

**Resources:** Tesfa Mekonen.

**Software:** Tesfa Mekonen.

**Supervision:** Gary C. K. Chan, Leanne Hides, Janni Leung.

**Validation:** Gary C. K. Chan, Lily Davidson, Leanne Hides, Janni Leung.

**Visualization:** Tesfa Mekonen, Lily Davidson, Janni Leung.

**Writing – original draft:** Tesfa Mekonen, Janni Leung.

**Writing – review & editing:** Tesfa Mekonen, Gary C. K. Chan, Tilahun Belete, Melak Menberu, Lily Davidson, Leanne Hides, Janni Leung.

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
