## [Decision Letter · Decision Letter 0]

17 Oct 2022

PONE-D-22-25327

Mental health service utilization in a low resource setting: A qualitative study on perspectives of health professionals in Northwest Ethiopia

PLOS ONE

Dear Dr. Mekonen,

Thank you for submitting your manuscript to PLOS ONE. After careful consideration, we feel that it has merit but does not fully meet PLOS ONE’s publication criteria as it currently stands. Therefore, we invite you to submit a revised version of the manuscript that addresses the points raised during the review process.

Dear Authors,

I have now received the review reports by two Reviewers. Both find merit in your manuscript and appreciate the significance of the topic.

However, there is a need to insert the findings into the Ethiopian contexts of education and practice, as suggested by Reviewer 2 and provide a thorough proof-reading/editing task, as suggested by Reviewer 1.

<ul> <li> 

 </li> <li> 

 </li> <li> 

 </li></ul>

We look forward to receiving your revised manuscript.

Kind regards,

Eleni Petkari

Academic Editor

PLOS ONE

Journal Requirements:

"We would like to thank the study participants for their involvement in this research. TM is supported by The University of Queensland Research and Training Program (RTP) scholarship. JL is supported by NHMRC fellowship. The data collection was financial supported by College of Medicine and Health Sciences, Bahir Dar University."

"This research did not receive any specific grant from funding agencies. The data collection cost was covered by College of Medicine and Health Sciences, Bahir Dar University, Ethiopia. The funder had no role in the study design, data collection and analysis, decision to publish, or preparation of the manuscript."

Reviewers' comments:

Reviewer's Responses to Questions

**Comments to the Author**

1. Is the manuscript technically sound, and do the data support the conclusions?

Reviewer #1: Yes

Reviewer #2: Yes

2. Has the statistical analysis been performed appropriately and rigorously? 

Reviewer #1: N/A

Reviewer #2: Yes

3. Have the authors made all data underlying the findings in their manuscript fully available?

Reviewer #1: Yes

Reviewer #2: No

4. Is the manuscript presented in an intelligible fashion and written in standard English?

Reviewer #1: Yes

Reviewer #2: Yes

5. Review Comments to the Author

Reviewer #1: The manuscript is written in standard manner, following the structure of a qualitative research. The themes were presented and interpreted well. The conclusions were corresponding to the obtained results.

Although the qualitative findings are very obvious (have not found any new insights), the significance of this research lies in the fact that the studied sample is underrepresented in such research areas, therefore, adds a lot to the body of literature.

The qualitative analytic approach was presented well.

The manuscript is written in comprehensive and academic manner.

Please edit for typos.

Reviewer #2: I congratulate the authors on conducting this much needed qualitative research amongst health professionals. However I have a few questions:

1. The sampling technique was purposive sampling. Why was such a small sample size selected? How many number of health professionals were approached for the interviews? Did they all agree?

2. Were there any differences between the responses of those health professionals seeing mental health issues in clinical practice compared to those who were not? Were there any differences between responses of nurses and doctors?

3. During various health professionals training, approximately how much time is spent on mental health training (theoretical and practical) in undergraduate and post graduate medical curriculum in Ethiopia? This fact could be included in the introduction and /or discussion while outlining recommendations

4. Was there any effect on prevalence of mental health issues in Ethiopia during the pandemic? An increase, if any, would reiterate the importance of this paper and statistics regarding same maybe included in the review of the literature.

5. Has a similar study been conducted in Ethiopia before? Kindly mention the same in the discussion

6. PLOS authors have the option to publish the peer review history of their article (what does this mean?). If published, this will include your full peer review and any attached files.

Reviewer #1: No

Reviewer #2: **Yes: **Dr Preeti M Galagali

---

## [Author Response · Author response to Decision Letter 0]

2 Nov 2022

Thank you for your comments and suggestions, which have considerably improved the quality of our manuscript. We have been able to address each of your points. Please find our point-by-point response, with page and line number cross-reference to changes in the manuscript below. A separate response to reviewers comment file is attached.

Editor: 

Journal Requirements:

[Response]: Thank you, the manuscript is formatted according to the journal requirements.

[Response]: Thank you, the grant information is provided in the cover letter.

"We would like to thank the study participants for their involvement in this research. TM is supported by The University of Queensland Research and Training Program (RTP) scholarship. JL is supported by NHMRC fellowship. The data collection was financial supported by College of Medicine and Health Sciences, Bahir Dar University."

"This research did not receive any specific grant from funding agencies. The data collection cost was covered by College of Medicine and Health Sciences, Bahir Dar University, Ethiopia. The funder had no role in the study design, data collection and analysis, decision to publish, or preparation of the manuscript."

[Response]: Thank you, we removed any funding related text from the manuscript and the amended statement is provided in the cover letter.

"This research did not receive any specific grant from funding agencies. The data collection cost was covered by College of Medicine and Health Sciences, Bahir Dar University, Ethiopia. TM is supported by The University of Queensland Research and Training Program (RTP) scholarship. The funder had no role in the study design, data collection and analysis, decision to publish, or preparation of the manuscript."

[Response]: Thank you for the detail information on the data availability. As the participants were relatively small number of health professionals, in a specific location, data contains potentially sensitive information to share it publicly. We have provided the data availability statement in the cover letter.

[Response]: Thank you, the caption for the supporting information is included at the end of the manuscript (page 23, Line 607 & 608). 

[Response]: Thank you, all the reference list is reviewed, and we ensured that it is correct and complete.

Reviewer: #1

The manuscript is written in standard manner, following the structure of a qualitative research. The themes were presented and interpreted well. The conclusions were corresponding to the obtained results.

Although the qualitative findings are very obvious (have not found any new insights), the significance of this research lies in the fact that the studied sample is underrepresented in such research areas, therefore, adds a lot to the body of literature.

The qualitative analytic approach was presented well.

The manuscript is written in comprehensive and academic manner.

Please edit for typos.

[Response]: Thank you for your positive assessment on the significance of the paper and quality design.

We did a thorough proof-reading/editing for the manuscript.

Reviewer: #2

I congratulate the authors on conducting this much needed qualitative research amongst health professionals. However I have a few questions:

[Response]: Thank you for your positive assessment on the importance of the paper.

1. The sampling technique was purposive sampling. Why was such a small sample size selected? How many number of health professionals were approached for the interviews? Did they all agree?

[Response]: We have used purposive sampling to include participants (key informants) with rich information and experience about our topic. We agree with the reviewer that the sample size looks small. However, as the participants were relatively homogeneous (all are health professionals) with respect to our topic, we have collected adequate qualitative data from this sample. We also used suitable qualitative analysis approach for a relatively small sample size. Previous qualitative studies in Ethiopia conducted in variety of mental health topics reported adequate qualitative data with similar sample size (e.g., Dadi et. al. 2021 – -- 13 health professionals, Yitbarek et.al. 2021 – -- 8 community health workers, and Hanlon et.al. 2017 – -- 17 health policy makers and leaders).

Regarding the health professionals approached, 17 health professionals were approached, 16 agreed and involved in the interview, one agreed to give interview but declined later citing busy schedule as a reason.

We updated the manuscript indicating that there was one participant who declined to involve in the interview (Page 5, Line 119 – 121).

“A total of 17 participants were approached and 16 (5 female,11 male) participants have participated in this study. One participant (male) agreed to be involved in the study but later declined the interview due to time constraints.”

2. Were there any differences between the responses of those health professionals seeing mental health issues in clinical practice compared to those who were not? Were there any differences between responses of nurses and doctors?

[Response]: We thank the reviewer for bringing this important issue. There were some differences in the participants response specially in Theme 1 “Avoidance of mental healthcare” and Theme 5 “Lack of government concern about mental health services”. In Theme 1, participants working in health centers (mostly nurses and public health officers) perceived that health centers are not the appropriate place to treat mental illness, whereas participants working in the hospitals (mostly mental health professionals and specialist doctors) perceived patient with mental illness should be treated at health center level (page 10, Line 209 – 216). 

“Health centers are perceived not to be the appropriate place to treat mental illness. This idea was exemplified by some of the participants working in the health centers (mostly nurses and public health officers) ….”

“Contrary to participants working in the health centers, participants working in hospitals (mostly mental health professionals and specialist doctors) implied that people with mental illness should be treated at health centers (e.g., patients that supposed to be treated at health centers are referred to specialized hospitals which creates undue burden [KI 3])”.

In addition, participants who had on-the-job training on mental health were more engaged in mental health service provision (Page 11, Line 231 – 234). 

“Despite the general tendency of health professionals to avoid providing mental healthcare, some participants who had on-the-job training on mental health reported more engagement in providing mental health services (e.g., I’m providing mental health treatment with full confidence after receiving the on-the-job training [KI 2).”

3. During various health professionals training, approximately how much time is spent on mental health training (theoretical and practical) in undergraduate and post graduate medical curriculum in Ethiopia? This fact could be included in the introduction and /or discussion while outlining recommendations

[Response]: Thank you. We have included the time spent in mental health training in Ethiopian Universities (Page 17/18 Line 395 – 401).

“Our result highlighted that mental health courses taught at universities do not adequately prepare graduates to provide mental health services. Medical students in Ethiopia spend approximately eight weeks on theoretical training and two to four weeks on practical training in mental health. Other health sciences students (e.g., Midwives and Nurses) spend approximately four weeks on theoretical and two weeks on practical training in mental health.”

4. Was there any effect on prevalence of mental health issues in Ethiopia during the pandemic? An increase, if any, would reiterate the importance of this paper and statistics regarding same maybe included in the review of the literature.

[Response]: Thank you, there were an increase in in the prevalence of mental illnesses such as depression during the pandemic. We have added this information in the background section of the manuscript (Page 3, Line 53 – 56)

“Due to the COVID-19 pandemic, mental illness has posed a great concern with a 25% worldwide increase in the prevalence of depression and anxiety [WHO, 2022]. In Ethiopia, a 3-fold increase in the prevalence of depression was reported compared to the pre-pandemic estimate [UN, 2020].”

5. Has a similar study been conducted in Ethiopia before? Kindly mention the same in the discussion

[Response]: Thank you. As we mentioned in the background section (Page 4, Line 74-76), to our knowledge, “there is a lack of research on the perspectives of health professionals on mental health services in Ethiopia”. We have included similar studies from Ethiopia in our discussion. 

Page 18, Line 409 – 410:

“Previous study in Ethiopia also reported that 98% of depression cases were undetected by primary care clinicians.”

Page 18, Line 412 – 415:

“Consistent with the results of the current study, previous studies reported the common perceptions among primary care clinicians that primary care facilities are not suited to treat mental illness and that specialists should treat mental illness”

Page 18/19, Line 425 – 427:

“A previous Ethiopian study also reported consistent finding that 88% of patients with mental illness visited religious and traditional places before seeking professional help.”

Page 19, Line 431 – 434:

“There are also reports from previous studies in Ethiopia that patients and family members expressed comfort in taking both pharmacological and religious/traditional treatments at the same time.”

---

## [Editor Report · Decision Letter 1]

8 Nov 2022

PONE-D-22-25327R1Mental health service utilization in a low resource setting: A qualitative study on perspectives of health professionals in Northwest EthiopiaPLOS ONE

Dear Dr. Mekonen,

Thank you for submitting your manuscript to PLOS ONE. After careful consideration, we feel that it has merit but does not fully meet PLOS ONE’s publication criteria as it currently stands. Therefore, we invite you to submit a revised version of the manuscript that addresses the points raised during the review process.

Dear Authors,

thank you so much for submitting a revised version of your manuscript, which now reads well and has addressed the Reviewers' concerns adequately.

Before the manuscript can be considered eligible for publication, please

1. Upload as a supplementary file a checklist explaining how your study addresses and fulfills the criteria for qualitative studies, using one of the following checklists:

the Consolidated criteria for reporting qualitative research (COREQ) checklist or Standards for reporting qualitative research (SRQR) checklist.

Further reporting guidelines can be found in the Equator Network's Guidelines for reporting qualitative research.

You can find the links to the above here, under the heading of Qualitative Studies: https://journals.plos.org/plosone/s/submission-guidelines#loc-qualitative-research

2. Cross-check again the language used to make sure the manuscript does not include fundamental orthographic and expression errors.

I would like to thank you for your timely submission of revisions and looking forward to receiving your amended manuscript.

We look forward to receiving your revised manuscript.

Kind regards,

Eleni Petkari

Academic Editor

PLOS ONE
---

## [Author Response · Author response to Decision Letter 1]

8 Nov 2022

Dear Editor,

Thank you for your comments and suggestions. We have revised the manuscript accordingly and we provided the Standards for reporting qualitative research (SRQR) checklist supplementary file. 

Editor: 

1. Upload as a supplementary file a checklist explaining how your study addresses and fulfills the criteria for qualitative studies, using one of the following checklists:

the Consolidated criteria for reporting qualitative research (COREQ) checklist or Standards for reporting qualitative research (SRQR) checklist.

Further reporting guidelines can be found in the Equator Network's Guidelines for reporting qualitative research.

You can find the links to the above here, under the heading of Qualitative Studies: https://journals.plos.org/plosone/s/submission-guidelines#loc-qualitative-research

[Response]: Thank you, we used the Standards for reporting qualitative research (SRQR) to guide our writing (Page 8, Line 181 & 182). The Standards for reporting qualitative research (SRQR) checklist is uploaded as supplementary file. The caption for the supplementary file is also included at the end of the manuscript (Page 23).

2. Cross-check again the language used to make sure the manuscript does not include fundamental orthographic and expression errors.

[Response]: Thank you, the manuscript is proofread for potential errors, revised, and edited. We provided the track changes of the edits in the marked-up copy of the manuscript

Thank you!

---

## [Editor Report · Decision Letter 2]

10 Nov 2022

Mental health service utilization in a low resource setting: A qualitative study on perspectives of health professionals in Northwest Ethiopia

PONE-D-22-25327R2

Dear Dr. Mekonen,

We’re pleased to inform you that your manuscript has been judged scientifically suitable for publication and will be formally accepted for publication once it meets all outstanding technical requirements.

Kind regards,

Eleni Petkari

Academic Editor

PLOS ONE
---

## [Editor Report · Acceptance letter]

15 Nov 2022

PONE-D-22-25327R2 

Mental health service utilization in a low resource setting: A qualitative study on perspectives of health professionals in Northwest Ethiopia 

Dear Dr. Mekonen:

I'm pleased to inform you that your manuscript has been deemed suitable for publication in PLOS ONE. Congratulations! Your manuscript is now with our production department. 

Kind regards, 

on behalf of

Dr. Eleni Petkari 

Academic Editor

PLOS ONE